# EDGE: KNOWLEDGE-DRIVEN NEW DRUG RECOMMENDATION

## ABSTRACT

Drug recommendation assists doctors in prescribing personalized medications to patients based on their health conditions. Existing drug recommendation solutions adopt the supervised multi-label classification setup and only work with existing drugs with sufficient prescription data from many patients. However, newly approved drugs do not have much historical prescription data and cannot leverage existing drug recommendation methods. To address this, we formulate the new drug recommendation as a few-shot learning problem. Yet, directly applying existing few-shot learning algorithms faces two challenges: (1) complex relations among diseases and drugs and (2) numerous false-negative patients who were eligible but did not yet use the new drugs. To tackle these challenges, we propose EDGE, which can quickly adapt to the recommendation for a new drug with limited prescription data from a few support patients. EDGE maintains a drug-dependent multi-phenotype few-shot learner to bridge the gap between existing and new drugs. Specifically, EDGE leverages the drug ontology to link new drugs to existing drugs with similar treatment effects and learns ontology-based drug representations. Such drug representations are used to customize the metric space of the phenotype-driven patient representations, which are composed of a set of phenotypes capturing complex patient health status. Lastly, EDGE eliminates the false-negative supervision signal using an external drug-disease knowledge base. We evaluate EDGE on two real-world datasets: the public EHR data (MIMIC-IV) and private industrial claims data. Results show that EDGE achieves 7.3% improvement on the ROC-AUC score over the best baseline.

## 1 INTRODUCTION

With the wide adoption of electronic health records (EHR) and the advance of deep learning models, we have seen great opportunities in assisting clinical decisions with deep learning models to improve resource utilization, healthcare quality, and patient safety (Xiao et al., 2018). Drug recommendation is one of the essential applications which aims at assisting doctors in recommending personalized medications to patients based on their health conditions. Existing drug recommendation methods typically formulate it as a supervised multi-label classification problem (Zhang et al., 2017; Zitnik et al., 2018; Shang et al., 2019b; Yang et al., 2021; Rui Wu & Wu., 2022; Tan et al., 2022b). They often train on massive prescription data to learn patient representations and use the learned representations to predict medications (i.e., labels). However, in reality, new drugs come to the market all the time. For example, U.S. Food and Drug Administration (FDA) approves a wide range of new drugs every year (FDA, 2022). Most of these newly approved drugs do not have much historical data to support model training (Blass, 2021). Even if sufficient prescription data for new drugs exists, existing models must be periodically re-trained or updated to recommend new drugs, which is expensive and complex. As a result, existing drug recommendation methods can only recommend the same set of drugs seen during training and are no longer applicable when new drugs appear.

To address this, we formulate the recommendation of new drugs as a few-shot classification problem. Given a new drug with limited prescription data from a few support patients (e.g., from clinical trials (Duijnhoven et al., 2013)), the model should quickly adapt to the recommendation for this drug. Meta-learning approaches have been widely used in such problems by learning how to quickly adapt the classifier to a new label unseen during training, given only a few support examples (Finn et al., 2017; Snell et al., 2017). However, most prior meta-learning works focus on vision or language-

related tasks. In the new drug recommendation, applying existing meta-learning algorithms faces the following challenges. **(1) Complex relations among diseases and drugs:** diseases and medicines can have inherent and higher order relations. Deciding whether to prescribe a drug to a specific patient depends on many factors, such as disease progression, comorbidities, ongoing treatments, individual drug response, and drug side effects. General meta-learning algorithms do not explicitly capture such dependencies. **(2) Numerous false-negative patients:** many drugs can treat the same disease, but usually, only one of them is prescribed. For any given drug, there exist many false-negative patients who were eligible but did not yet use the new drug (e.g., due to drug availability, doctor's preference, or insurance coverage). The number of false-negative supervision signals will substantially confuse the model learning, especially in the few-shot learning setting.

To address these challenges, we introduce `EDGE`, a drug-dependent multi-phenotype few-shot learner to quickly adapt to the recommendation for a new drug with limited support patients. Specifically, since drugs within the same category often have similar treatment effects, `EDGE` utilizes the drug ontology for drug representation learning to link new drugs with existing drugs. Further, `EDGE` learns multi-phenotype patient representations to capture the complex patient health status from different aspects such as chronic diseases, current symptoms, and ongoing treatments. Given a new drug with a few support patients, `EDGE` makes recommendations by performing a drug-dependent phenotype-level comparison between representations of query patients and corresponding support prototypes. Lastly, to reduce the false-negative supervision signal, `EDGE` leverages the MEDI (Wei et al., 2013) drug-disease knowledge base to guide the negative sampling process.

The **main contributions** of this work include:

- To our best knowledge, this is the first work formulating the task of new drug recommendation;
- We propose a meta-learning framework `EDGE` to solve this problem by considering complicated relations among diseases and drugs, and eliminating numerous false-negative patients.
- We conduct extensive experiments on the public EHR data MIMIC-IV (Johnson et al., 2020) and private industrial claims data. Results show that our approach achieves 5.6% over ROC-AUC, 6.3% over Precision@100, and 5.5% over Recall@100 when providing recommended patient lists for new drugs. We also include detailed analyses and ablation studies to show the effectiveness of multi-phenotype patient representation, drug-dependent patient distance, and knowledge-guided negative sampling.

## 2 PROBLEM FORMULATION AND PRELIMINARIES

Denote the set of all drugs as $\mathcal{M}$; the goal of drug recommendation is to prescribe drugs in $\mathcal{M}$ that are suitable for a patient with a record $v = [c_1, \ldots, c_V]$, which consists of a list of diseases (and procedures)[1], and $V$ is the total number of diseases and procedures in the record $v$. Prior works (Zhang et al., 2017; Shang et al., 2019b; Yang et al., 2021; Tan et al., 2022b) formulate drug recommendation as a multi-label classification problem by generating a multi-hot output of size $|\mathcal{M}|$. However, this formulation assumes that the drug label space $\mathcal{M}$ remains unchanged after training and is not applicable when new drugs appear. Thus, we propose an alternative formulation for the new drug recommendation as follows.

Assume the entire drug set $\mathcal{M}$ is partitioned into a set of existing drugs $\mathcal{M}^{old}$ and a set of new drugs $\mathcal{M}^{new}$, where $\mathcal{M}^{old} \cap \mathcal{M}^{new} = \emptyset$. Each existing drug $m_i \in \mathcal{M}^{old}$ has sufficient patients using the drug $m_i$ (e.g., from EHR data). Each new drug $m_t \in \mathcal{M}^{new}$ is associated with a small support set $\mathcal{S}_t = \{v_j\}_{j=1}^{N_s}$ consisting of patients using the drug $m_t$ (e.g., from clinical trials), and an unlabeled query patient set $\mathcal{Q}_t = \{v_j\}_{j=1}^{N_q}$, where $N_s$ and $N_q$ are the number of patients in the support and query sets, respectively. The goal of new drug recommendation is to train a model $f_\phi(\cdot)$ parameterized by $\phi$ on existing drugs $\mathcal{M}^{old}$, such that it can adapt to new drug $m_t \in \mathcal{M}^{new}$ given the small support set $\mathcal{S}_t$, and make correct recommendation on the query set $\mathcal{Q}_t$.

Our work is inspired by the prototypical network (Snell et al., 2017), which learns a representation model $f_\phi(\cdot)$ such that patients using a specific drug will cluster around a prototype representation. Recommendation can then be performed by computing the distance to the prototype. To equip the

---

[1]To reduce clutter, we use a unified notation for both diseases and procedures. Since we focus on record-level prediction, "patient" and "record" are used interchangeably.

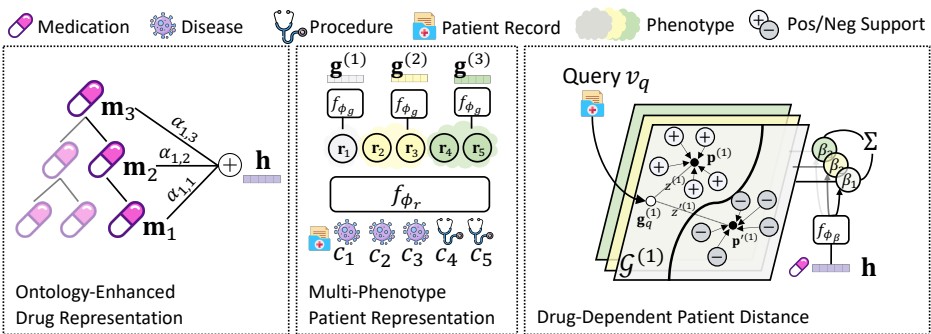

Figure 1: EDGE learns the ontology-enhanced drug representation $\mathbf{h}$ and multi-phenotype patient representations $\{\mathbf{g}^{(l)}\}_{l=1}^{3}$. For a new drug $m_1$, EDGE decides whether to prescribe it to a query patient $v_q$ by performing a drug-dependent phenotype-level comparison between multi-phenotype query representations $\{\mathbf{g}_q^{(l)}\}_{l=1}^{3}$ and corresponding support prototypes $\{\mathbf{p}^{(l)}\}_{l=1}^{3}$.

model with the ability to adapt to new drug with limited support patients, prototypical network trains the model via episodic training, where each episode is designed to mimic the low-data testing regime. Concretely, an episode is formed by first sampling an existing drug $m_i$ from $\mathcal{M}^{old}$ and then sampling a set of patients using the drug $m_i$. The sampled patients are divided into two disjoint sets: (1) a support set $\mathcal{S}_i$ used to calculate the prototype, and (2) a query set $\mathcal{Q}_i$ used to calculate the loss. From the support set $\mathcal{S}_i$, prototypical network calculates the prototype representation as,

$$\mathbf{p} = \frac{1}{|\mathcal{S}_i|} \sum_{j \in \mathcal{S}_i} f_\phi(v_j), \ \ \mathbf{p} \in \mathbb{R}^e, \tag{1}$$

where $\mathbf{p}$ is an $e$-dimensional vector in the metric space, and $|\cdot|$ denotes cardinality. Next, given a query patient $v_q$, the probability of recommending drug $m_i$ is measured by the distance $d(\cdot)$ between its representation and the corresponding prototypes as,

$$p_\phi(y_q = +|v_q) = \frac{\exp\left(-d\left(f_\phi\left(v_q\right), \mathbf{p}\right)\right)}{\exp\left(-d\left(f_\phi\left(v_q\right), \mathbf{p}\right)\right) + \exp\left(-d\left(f_\phi\left(v_q\right), \mathbf{p}'\right)\right)}, \tag{2}$$

where $\mathbf{p}'$ is the negative prototype obtained from another negative support set $\mathcal{S}_i'$ of patients not using the drug $m_i$ (i.e., negative sampling). The loss is computed as the negative log-likelihood (NLL) loss $\mathcal{L}(\phi) = -\log p_\phi(y_q = *|v_q)$ of the true label $* \in \{+, -\}$. And the model $f_\phi(\cdot)$ is optimized on both the query set $\mathcal{Q}_i$ and another negative query set $\mathcal{Q}_i'$ obtained via negative sampling (similarly as $\mathcal{S}_i'$).

## 3 KNOWLEDGE-DRIVEN NEW DRUG RECOMMENDATION

In this section, we introduce EDGE, which can adapt to new drugs with limited support patients via a drug-dependent multi-phenotype few-shot learner. Specifically, EDGE consists of the following modules: **(1) Ontology-enhanced drug encoder** that fuses ontology information into drug representation to link new drugs to existing drugs with similar treatment effects; **(2) Multi-phenotype patient encoder** that represents each patient with a set of phenotype-level representations to capture the complex patient's health status; **(3) Drug-dependent distance measures** that learns drug-dependent phenotype importance scores to customize the patient similarity; **(4) knowledge-guide negative sampling** that eliminates the false-negative supervision signal. Figure 1 provides an illustration of EDGE. In the following, we will describe how EDGE decides whether to prescribe a drug $m_i$ to a query patient $v_q$, given a small set of support patients $\mathcal{S}_i$ using the drug $m_i$.

### 3.1 ONTOLOGY-ENHANCED DRUG REPRESENTATION LEARNING

Though many new drugs have not been used regularly in clinical practice, they still belong to the same drug category (from a drug ontology) as some existing drugs and share similar treatment effects, implicitly indicating similar patient populations. For example, the newly approved *Quviviq* for treating insomnia belongs to the same category (*Orexin Receptor Antagonist*) as some existing

drugs, like *Belsomra* and *Dayvigo*, which are also sleeping aids. We here leverage the drug ontology to enrich the drug representation by attentively combing the drug itself and its corresponding ancestors (e.g., higher-level drug categories).

Concretely, for the drug $m_i$, we obtain its basic embeddings $\mathbf{m}_i \in \mathbb{R}^e$ by feeding its description[2] into Clinical-BERT (Alsentzer et al., 2019). Then, follow Choi et al. (2017), we use the basic embeddings of drug $m_i$ and its ancestors to calculate the ontology-enriched drug representation as,

$$\mathbf{h} = \sum_{j \in \mathcal{A}_i} \alpha_{i,j} \mathbf{m}_j, \ \ \mathbf{h} \in \mathbb{R}^e, \tag{3}$$

where $\mathcal{A}_i$ denotes the set of drug $m_i$ and its ancestors, and the attention score $\alpha_{i,j}$ represents the importance of ancestor $m_j$ for drug $m_i$, which is calculated as,

$$\alpha_{i,j} = \frac{\exp(f_{\phi_a}(\mathbf{m}_i \oplus \mathbf{m}_j))}{\sum_{k \in \mathcal{A}_i} \exp(f_{\phi_a}(\mathbf{m}_i \oplus \mathbf{m}_k))}, \ \ \alpha_{i,j} \in [0, 1], \tag{4}$$

where $\oplus$ denotes the concatenation operator, and $f_{\phi_a}(\cdot) : \mathbb{R}^{2e} \mapsto \mathbb{R}$ is defined as a two-layer fully connected neural network with Tanh activation. In this way, we fuse the ontology information into the representation $\mathbf{h}$ for drug $m_i$, which is later used to customize the metric space of phenotype-driven patient representations, introduced next.

## 3.2 Multi-phenotype patient representation learning

Patient health status includes many factors, such as disease progression, comorbidities, ongoing treatments, individual drug response, and drug side effects. Encoding each patient into a single vector may not capture the complete information, especially for patients with complex health conditions. Therefore, we define a set of phenotypes and represent each patient with a set of phenotype vectors. Each phenotype can provide helpful guidance in patient representation learning and further benefit the new drug recommendation.

Specifically, for every support/query patient $v$ with a list of diseases $[c_1, \ldots, c_V]$, EDGE first computes the contextualized disease representations by applying the embedding function $f_{\phi_r}(\cdot)$ as,

$$[\mathbf{r}_1, \ldots, \mathbf{r}_V] = f_{\phi_r}([c_1, \ldots, c_V]), \ \ \mathbf{r}_j \in \mathbb{R}^e, \tag{5}$$

where $\mathbf{r}_j$ is the contextualized representation for disease $c_j$. We model $f_{\phi_r}(\cdot)$ using a bi-directional gated recurrent unit (GRU) due to its popularity in prior works (Zhang et al., 2017; Shang et al., 2019b; Yang et al., 2021), and also show results with multilayer perceptron (MLP) and Transformer (Vaswani et al., 2017) in our experiments.

Next, we leverage domain knowledge to group diseases into different phenotypes. To obtain the representation for the $l$-th phenotype, we take the representations from all diseases that belong to that phenotype, project them to a lower dimension, and calculate their mean representations as,

$$\mathbf{g}^{(l)} = \frac{1}{|\mathcal{G}^{(l)}|} \sum_{j \in \mathcal{G}^{(l)}} f_{\phi_g}(\mathbf{r}_j), \ \ \mathbf{g}^{(l)} \in \mathbb{R}^g, \tag{6}$$

where $\mathcal{G}^{(l)}$ represents the set of diseases whose phenotype is $l$, and $f_{\phi_g}(\cdot) : \mathbb{R}^e \to \mathbb{R}^g$ is single-layer neural network and $g < e$. We show results with different values of $g$ in the experiment. If $\mathcal{G}^{(l)}$ is empty, we take the pooled sequence representation as a substitute. The phenotypes are extracted from Clinical Classification Software (CCS) (H. CUP, 2010). There are 511 phenotypes in total. In this way, each support/query patient is represented with a set of phenotype vectors $\{\mathbf{g}^{(l)}\}_{l=1}^L$.

Based on the multi-phenotype patient representations, we further calculate the phenotype-level prototypes from the support set $\mathcal{S}_i$ of drug $m_i$, where equation 1 is revised as,

$$\mathbf{p}^{(l)} = \frac{1}{|\mathcal{S}_i|} \sum_{j \in \mathcal{S}_i} \mathbf{g}_j^{(l)}, \ \ \mathbf{p}^{(l)} \in \mathbb{R}^g, \tag{7}$$

where $\mathbf{g}_j^{(l)}$ is the $l$-th phenotype representation for patient $v_j$ from the support set $\mathcal{S}_i$ of drug $m_i$. In this way, we encode the support set $\mathcal{S}_i$ into a set of phenotype-level prototypes $\{\mathbf{p}^{(l)}\}_{l=1}^L$, which is further used to calculate the drug-dependent patient distance with the multi-phenotype representations $\{\mathbf{g}_q^{(l)}\}_{l=1}^L$ of the query patient $v_q$, described next.

---

[2]E.g., *Ibuprofen is a nonsteroidal anti-inflammatory drug that is used for treating pain, fever, and inflammation.*

### 3.3 DRUG-DEPENDENT PATIENT DISTANCE

We finally focus on defining a reasonable metric to measure the distance between query patient and support prototypes. Intuitively, different drugs may have different focuses when comparing patients. For example, chronic drugs may emphasize chronic diseases and patients' past health history, while drugs accompanying specific treatments may focus on ongoing treatments. Therefore, we leverage the drug representation to learn a drug-dependent patient distance.

Specifically, given the phenotype-driven prototypes $\{\mathbf{p}^{(l)}\}_{l=1}^L$ from support patients $\mathcal{S}_i$, and the multi-phenotype representations $\{\mathbf{g}_q^{(l)}\}_{l=1}^L$ for the query patient $v_q$, we first calculate the per-phenotype distance between the query and support. Formally, for a specific phenotype $l$, the distance is defined as,

$$z^{(l)} = d(\mathbf{g}_q^{(l)}, \mathbf{p}^{(l)}), \ \ z^{(l)} \in \mathbb{R}, \tag{8}$$

where $d(\cdot)$ is the euclidean distance. We also show results with cosine distance in the experiment.

---

**Algorithm 1** Training for EDGE

**Require:** $\mathcal{M}^{old}$: existing training drugs; $\mathcal{A}$: drug ontology; $\{\mathcal{G}^{(l)}\}_{l=1}^L$: phenotypes
1: **while** not done **do**
2:     Sample a drug $m_i$ from $\mathcal{M}^{old}$
3:     Sample support and query set $\mathcal{S}_i$, $\mathcal{Q}_i$
4:     Get drug representation $\mathbf{h}$ with Eq. 3
5:     Calculate phenotype-driven prototypes $\{\mathbf{p}^{(l)}\}_{l=1}^L$ for support $\mathcal{S}_i$ with Eq. 5, 6, 7
6:     **for all** $v_q \in \mathcal{Q}_i$ **do**
7:         Compute multi-phenotype patient representations $\{\mathbf{g}_q^{(l)}\}_{l=1}^L$ for $v_q$ with Eq. 5, 6
8:         Compute drug-dependent phenotype importance $\boldsymbol{\beta}$ with Eq. 9
9:         Get per-phenotype distances $\mathbf{z}$ between support and query with Eq. 8
10:         Get recommendation with Eq. 10
11:     **end for**
12:     Use gradient descent to update parameters $\phi$ based on the NLL loss
13: **end while**

---

In practice, we only calculate the distances between the union of phenotypes that appear in both query and support patients and mask all other distances as zero. Combining all $L$ per-phenotype distances, we obtain a $L$-dimensional distance vector, denoted as $\mathbf{z}$.

Next, with the ontology-enhanced drug representation $\mathbf{h}$ for drug $m_i$, we calculate the drug-dependent importance weights over different phenotypes as,

$$\boldsymbol{\beta} = \sigma(f_{\phi_\beta}(\mathbf{h})), \ \ \boldsymbol{\beta} \in [0,1]^L, \tag{9}$$

where $\sigma(\cdot)$ denotes the sigmoid function, and $f_{\phi_\beta}(\cdot) : \mathbb{R}^e \mapsto \mathbb{R}^L$ is a single-layer fully connected neural network. According to the importance weight, the probability of recommending drug $m_i$ to the query patient $v_q$ (equation 2) is reformulated as,

$$p_\phi(y_q = +|v_q) = \frac{\exp(-\boldsymbol{\beta}^\top \mathbf{z})}{\exp(-\boldsymbol{\beta}^\top \mathbf{z}) + \exp(-\boldsymbol{\beta}^\top \mathbf{z}')}, \tag{10}$$

where $\boldsymbol{\beta}^\top \mathbf{z}$ aggregates the per-phenotype distances $\mathbf{z}$ with respect to drug $m_i$ using the learned importance weights $\boldsymbol{\beta}$, and $\mathbf{z}'$ is the distance vector between the query $v_q$ and the negative (i.e., not using drug $m_i$) support patients obtained via negative sampling, introduced next.

### 3.4 KNOWLEDGE-GUIDED NEGATIVE SAMPLING

In addition to the model design, negative sampling plays a vital role in model training. Uniform negative sampling is the default method, where all patients not using a particular drug are treated as negatives. However, in real-world scenarios, many drugs can treat the same disease, but usually, only one of them is prescribed. For any given drug, there exist many false-negative patients that could but eventually did not use the drug for several reasons,

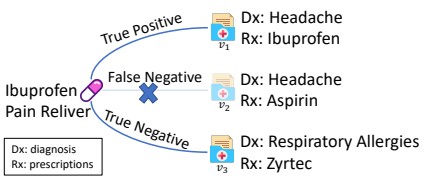

Figure 2: A false negative example.

such as availability, doctor preference, or insurance coverage. For example, in figure 2, for drug *Ibuprofen*, uniform negative sampling treats both the patients $v_2$ and $v_3$ as negatives since they are not prescribed with *Ibuprofen*. However, patient $v_2$ is false negative as *Ibuprofen* can also be used. We quantitatively evaluate the prevalence of false-negative patients in experiment, which ranges from 20% to 30%. The number of false-negative supervision signals will confuse the model learning, especially in the few-shot learning setting.

To address this issue, `EDGE` introduces the following knowledge-guided negative sampling strategy. We leverage the MEDI (Wei et al., 2013) drug-disease relationship (i.e., pairs of drugs and their target diseases ). When performing negative sampling for a given drug, we only sample from the negative patients whose diagnoses do not overlap with the listed target diseases of that drug. Note that we only use this strategy during training so that the information will not leak to testing. We empirically show that this simple strategy can improves the performance.

# 4 EXPERIMENTS

## 4.1 EXPERIMENTAL SETTINGS

**Dataset Description.** We evaluate `EDGE` on two real-world EHR datasets: MIMIC-IV and Claims, which are described as follows (see detailed data statistics and descriptions in appendix A.1):

- **MIMIC-IV** (Johnson et al., 2020) covers more than 382K patients admitted to the ICU or the emergency department at the Beth Israel Deaconess Medical Center between 2008 to 2019. We split the drugs by the time they were first prescribed. We use drugs first appeared in 2008 as the training (existing) drugs, drugs appeared in 2009 as the validation drugs, and drugs appeared after 2010 as the testing (new) drugs. The admissions are split accordingly: if an admission contains any test/validation/training drug, it will be in test/validation/training set. If an admission contains drugs from more than one split, priority is given to test, validation, and training. There are 49 new drugs in total.
- **Claims** are captured from payers and healthcare providers across the United States. We randomly sample 30K patients and select all their claims from 2015 to 2019, resulting in 691K claims. We split the drugs into 70%/10%/20% training/validation/test sets. The admissions are split the same way as MIMIC-IV. There are 99 new drugs in total.

**Baselines.** We compare `EDGE` to three categories of baselines. (1) The first category is **drug recommendation** methods, including RNN (Choi et al., 2016b), GAMENet (Shang et al., 2019b), SafeDrug (Yang et al., 2021). For a fair comparison, the drug recommendation is first trained on the training set of existing drugs, and then fine-tuned on the support set of new drugs. (2) The second category is **few-shot learning** approaches, including MAML (Finn et al., 2017), ProtoNet (Snell et al., 2017), FEAT (Ye et al., 2021), CTML (Peng & Pan, 2022). (3) Lastly, we include the **cold-start recommendation** methods, including MeLU (Lee et al., 2019) and TaNP (Lin et al., 2021). See appendix A.2 for details of all baselines.

**Evaluation Metrics.** We randomly generate 1000 episodes for the test drugs. Each episode contains a randomly sampled test drug, a support set consisting of 5 positive and 25 negative records for this drug, and a query set consisting of all the rest of testing records (over 15K for MIMIC-IV data and 29K for claims data). Each episode is a binary classification task: whether to prescribe the drug to query record or not. We calculate the Area Under Receiver Operator Characteristic Curve (ROC-AUC), Area Under Precision-Recall Curve (PR-AUC), Precision@K, Recall@K. For each metric, we also report the 95% confidence interval calculated from the 1000 episodes. We also perform the independent two-sample t-test to evaluate if `EDGE` achieves significant improvement over baseline methods. Due to space limitations, we only show ROC-AUC, Precision@100, and Recall@100 in the main paper and leave other metrics in the appendix B.1.

**Implementation Details.** We use ICD codes (WHO, 1993) to represent disease and procedure, and ATC-5 level codes (WHO, 1976) to represent medication. CCS category (H. CUP, 2010) is used to define the phenotype for diseases and procedures. ATC ontology WHO (1976) is used to build the drug ontology. We obtain the basic code embeddings by encoding code description with Clinical-BERT (Alsentzer et al., 2019). For all methods, we use the bi-directional GRU as the backbone encoder, with 768 as input dimension and 512 as the output dimension. We set the phenotype vector dimension $g$ as 64. All models are trained for 100K episodes with 5 positive and 250 negative supports. We select the best model by monitoring the ROC-AUC score on the validation set.

## 4.2 MAIN RESULTS

Table 1 evaluates `EDGE` on the performance of recommendations on new drugs. First, we observe that existing drug recommendation methods perform poorly for this task due to limited training examples. Among them, SafeDrug achieves slightly better performance, probably due to the usage of

Table 1: Results on new drug recommendation. $\pm$ denotes the 95% confidence interval. **Bold** indicates the best result and underline indicates second best. * indicates that EDGE achieves significant improvement over the best baseline method (i.e., the p-value is smaller than 0.05). Experiment results show that EDGE can adapt to new drugs and make correct recommendations.

| Method | MIMIC-IV | | | Claims | | |
| | ROC-AUC | Precision@100 | Recall@100 | ROC-AUC | Precision@100 | Recall@100 |
|---|---|---|---|---|---|---|
| RNN | $0.4612 \pm 0.0091$ | $0.0279 \pm 0.0062$ | $0.0040 \pm 0.0005$ | $0.5012 \pm 0.0062$ | $0.0130 \pm 0.0014$ | $0.0047 \pm 0.0006$ |
| GAMENet | $0.6831 \pm 0.0087$ | $0.1047 \pm 0.0121$ | $0.0722 \pm 0.0070$ | $0.6361 \pm 0.0077$ | $0.0338 \pm 0.0058$ | $0.0203 \pm 0.0044$ |
| SafeDrug | $0.7488 \pm 0.0102$ | $0.1055 \pm 0.0103$ | $0.0655 \pm 0.0064$ | $0.4792 \pm 0.0048$ | $0.0129 \pm 0.0016$ | $0.0022 \pm 0.0003$ |
| MAML | $0.7197 \pm 0.0105$ | $0.0549 \pm 0.0070$ | $0.0356 \pm 0.0045$ | $0.6019 \pm 0.0074$ | $0.0223 \pm 0.0044$ | $0.0086 \pm 0.0015$ |
| ProtoNet | $0.7903 \pm 0.0084$ | $0.1426 \pm 0.0116$ | $\underline{0.1187} \pm 0.0090$ | $\underline{0.6740} \pm 0.0070$ | $\underline{0.0812} \pm 0.0087$ | $\underline{0.0436} \pm 0.0053$ |
| FEAT | $\underline{0.8020} \pm 0.0081$ | $\underline{0.1427} \pm 0.0120$ | $0.1080 \pm 0.0077$ | $0.6709 \pm 0.0079$ | $0.0718 \pm 0.0101$ | $0.0349 \pm 0.0068$ |
| CTML | $0.5960 \pm 0.0113$ | $0.0985 \pm 0.0098$ | $0.0829 \pm 0.0072$ | $0.5550 \pm 0.0072$ | $0.0102 \pm 0.0014$ | $0.0049 \pm 0.0008$ |
| MeLU | $0.7070 \pm 0.0080$ | $0.0600 \pm 0.0089$ | $0.0355 \pm 0.0052$ | $0.5932 \pm 0.0068$ | $0.0264 \pm 0.0050$ | $0.0060 \pm 0.0009$ |
| TaNP | $0.6093 \pm 0.0093$ | $0.0243 \pm 0.0061$ | $0.0092 \pm 0.0019$ | $0.5608 \pm 0.0076$ | $0.0134 \pm 0.0021$ | $0.0067 \pm 0.0012$ |
| EDGE | $\mathbf{0.8608 \pm 0.0069}$* | $\mathbf{0.2251 \pm 0.0139}$* | $\mathbf{0.1907 \pm 0.0126}$* | $\mathbf{0.7275 \pm 0.0066}$* | $\mathbf{0.1254 \pm 0.0102}$* | $\mathbf{0.0803 \pm 0.0075}$* |

drug molecule information. General few-shot learning achieves inconsistent performance. Metric-based methods ProtoNet and FEAT performance better compared to optimization-based approaches MAML and CTML. This may be due to the challenge in imbalanced classification setting, where the number of negatives is much larger than that of positives, and the optimization-based approach might get trapped in the local minimum. Among general recommendation methods, MeLU and TaNP do not perform very well for this task despite their strong performance reported in Tan et al. (2022a). We suspect that they adapt the model based on patient history for rare disease diagnoses. While the setting is different in our case, we need to adapt the model based on the drug (i.e., item) prescription history. Lastly, EDGE achieves the best performance in all metrics on both datasets. Specifically, compared to the best baseline, on MIMIC-IV, EDGE achieves 5.9% absolute improvements on ROC-AUC, 8.2% on Precision@100, and 7.2% on Recall@100; for claims data, EDGE achieves 5.3% on ROC-AUC, 4.4% on Precision@100, and 3.7% on Recall@100.

## 4.3 QUANTITATIVE ANALYSIS OF PERFORMANCE GAINS

This section presents quantitative analyses to understand the performance gains of EDGE, including the analysis of false negatives, the ablation study of proposed modules, and sensitivity of the ratio of negative supports and the dimension of phenotype. We conduct additional analysis on different backbone models in appendix B.2 and analysis on different distance measures in appendix B.3.

**Analysis on False Negatives.** Due to the one-to-many mapping between disease and drug, there exist many false-negative patients for a given drug. For each drug, we calculate the prevalence of false-negative records (i.e., percentage of records that could but did not use the drug) using the MEDI (Wei et al., 2013) drug-disease database. We report the average prevalence across all new drugs in

Table 2: Analysis on the influence of false negatives. Prev.: prevalence; Orig.: original; Adj: adjusted.

| Dataset | Prev. | Orig. P@100 | Adj. P@100 |
|---|---|---|---|
| MIMIC-IV | 31.5% | 0.2251 | 0.4053 |
| Claims | 22.1% | 0.1254 | 0.3317 |

table 2. The false-negative records range from 20 to 30% for both datasets. This level of noise can largely confuse the model if not appropriately handled. We then correct the false-negative labels (i.e., swap them from negative to positive) in the test query set, and re-calculate the adjusted Precision@100 for EDGE, which increases by 18.0% on MIMIC-IV data and by 20.6% on claims data.

**Ablation Study of Module Importance.** In the ablation study, we evaluate the contribution of each proposed module. According to the results in table 3, we observe that the multi-phenotype patient encoder contributes the most: switching it with a single vector patient representation will decrease the ROC-AUC by 16.9%. Removing drug-dependent patient distance also decreases the performance by a large margin, as the model cannot adapt the metric space to a specific drug. The ontology-enhanced drug representation and knowledge-guided negative sampling both slightly improve the model performance. Changing ontology-enhanced drug representation to the basic drug representation obtained from the drug name and description ignores the high-level category information shared between existing and new drugs. Changing knowledge-guided negative sampling to uniform random sampling might confuse the model with the prevalent false negatives.

Table 3: Ablation study.

| Method | MIMIC-IV | | | Claims | | |
|---|---|---|---|---|---|---|
| | ROC-AUC | P@100 | R@100 | ROC-AUC | P@100 | R@100 |
| Remove ontology-enhanced drug representation | 0.8239 | 0.1584 | 0.1411 | 0.6809 | 0.0985 | 0.0672 |
| Remove multi-phenotype patient representation | 0.7992 | 0.1555 | 0.1312 | 0.6910 | 0.0818 | 0.0551 |
| Remove drug-dependent distance measure | 0.8033 | 0.1687 | 0.1542 | 0.7078 | 0.1055 | 0.0758 |
| Remove knowledge-guided negative sampling | 0.8175 | 0.1899 | 0.1725 | 0.7149 | 0.1116 | 0.0789 |
| EDGE | **0.8608** | **0.2251** | **0.1907** | **0.7275** | **0.1254** | **0.0803** |

**Analysis on the Influence of Hyper-parameters.** We evaluate the effect of two important hyper-parameters: the ratio of negative supports and the dimension of the phenotype vector on MIMIC-IV data. The results can be found in figure 3, where the results of both EDGE and FEAT (best baseline) are reported. The performance increases as the ratio of negative support records or the dimension of phenotype representation. Inter-

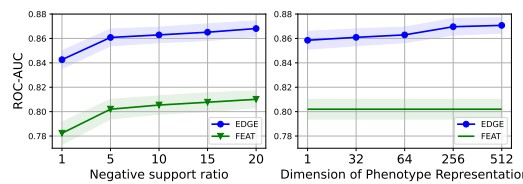

Figure 3: The effect of negative support ratio and dimension of phenotype representation.

estingly, we find the EDGE can outperform the best baseline method even if we set the dimension of phenotype to 1. That is to represent each patient with a single vector (the same as the prototypical network). This confirms the benefits of leveraging phenotype knowledge and drug ontology.

## 4.4 QUALITATIVE ANALYSIS

This section presents qualitative analysis to further investigate the performance of different drugs and the learned importance scores and representations.

**Analysis on Per-Category Performance.** We first group new drugs by ATC-2 level and calculate the per-category performance of EDGE on MIMIC-IV data. Due to space limitations, we report the top-5 and bottom-5 categories in table 4 and show the full results in the appendix B.4. From the per-category result, the best-performing drugs are typically more targeting specific diseases. For example, *antineoplastic agents* are medications used to treat *cancer*, and *antivirals* are drugs targeting specific virals such as *influenza* and *rabies*. The support patients for such drugs usually have specific diseases that act as a prominent indicator for the prescription of the drugs. On

Table 4: Performance by drug category.

| Drug Category | ROC-AUC |
|---|---|
| Antineoplastic agents | 0.9882 |
| Antivirals | 0.9754 |
| Urologicals | 0.9738 |
| Anti-parkinson drugs | 0.9611 |
| Other alimentary tract products | 0.9412 |
| ... | ... |
| Antibacterials | 0.7888 |
| All other therapeutic products | 0.7827 |
| Antiinfective | 0.7367 |
| Calcium channel blockers | 0.7285 |
| Beta-adrenergic blockers | 0.6074 |

the other hand, the worst performing drugs are usually very general, such as *antibacterials* or *anti-infectives*, which applies to various circumstances. The limited number of support patients cannot capture the broad applications of these drugs, which leads to lower performance.

**Analysis of the Importance of Phenotypes.** We show three example drugs from the testing set with the top-3 phenotypes ranked by the learned importance scores. The result can be found in table 5. The top-ranked phenotypes match well with the usage of each drug. For example, for *dactinomycin* which treats a variety of *cancers*, the top-ranked phenotypes are all cancer-related. Interestingly, EDGE also learns to pay attention to some common co-morbidities for a given disease. For example, for the drug *rasagiline* which is used for *Parkinson's disease*, EDGE also attends to *hypertension* and *anemia*, which are often seen in patients with *Parkinson's disease*.

Table 5: Top-3 phenotypes ranked by the learned importance scores.

| Drug | Usage | Top-3 Phenotypes |
|---|---|---|
| Dactinomycin | Cancer | Other cancer
Lung Cancer
Neoplasms |
| Nebivolol | Hypertension | Essential hypertension
Circulatory disease
Surgical procedures |
| Rasagiline | Parkinson | Parkinson
Hypertension
Anemia |

**Analysis on Drug Representation.** For qualitative analysis, we visualize the learned ontology-enhanced drug representation on MIMIC-IV data with t-SNE (van der Maaten & Hinton, 2008). As shown in figure 4, the representations for drugs with similar treatment effects cluster into smaller

categories. Further, the representations for new drugs also align with existing drugs. This indicates that the model adapts to new drugs by linking the new drugs to existing drugs with similar treatment effects.

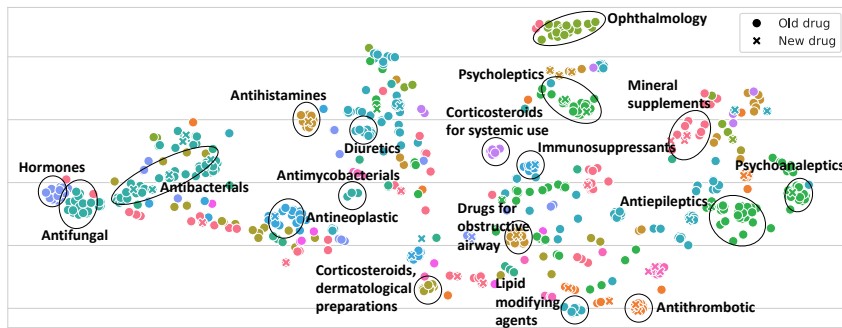

Figure 4: t-SNE plot of the learned ontology-enhanced drug representation. Node color represents ATC-2 level (coarse) categories while circle represents ATC-4 (fine-grained) level categories.

## 5    RELATED WORK

**Drug Recommendation.** Drug recommendation aims at suggesting personalized medications to patients based on their specific health conditions. In practice, drug recommendation can act as a decision support system that helps doctors by supporting decision-making and reducing workload. Existing works typically formulate the problem as a multi-label classification task. LEAP (Zhang et al., 2017) proposes a sequence-to-sequence model to predict drugs given the patient's diagnoses. Later works try to model the longitudinal relations via knowledge graph (Wang et al., 2017), attention mechanism (Choi et al., 2016a), model pre-training (Shang et al., 2019a), and memory network (Le et al., 2018; Shang et al., 2019b). More recent works further incorporate more domain-related inductive bias. For example, SafeDrug (Yang et al., 2021) leverages the global and local molecule information, and 4SDrug (Tan et al., 2022b) utilizes the set-to-set (set of symptoms to set of medications) module. Despite the good performance on existing drugs, these models are no longer applicable when new drugs appear. That is why we study new drug recommendation problem in this paper.

**Few-Shot Learning.** Few-shot learning aims at quickly generalizing the model to new tasks with a few labeled samples (Wang et al., 2020). Existing works can be categorized into metric-learning based approaches that aims to establish similarity or dissimilarity between classes (Vinyals et al., 2016; Sung et al., 2017; Snell et al., 2017; Oreshkin et al., 2019; Cao et al., 2021; Ye et al., 2021), and optimization based approach seeks to learn a good initialization point that can adapt to new tasks within a few parameter updates (Finn et al., 2017; Nichol et al., 2018; Rusu et al., 2019; Yao et al., 2021b; Peng & Pan, 2022). For recommendation problem, few-shot learning is leveraged to solve the cold-start problem, where the goal is to quickly adapt the model to new users with limited history (Lee et al., 2019; Du et al., 2022; Dong et al., 2020; Lin et al., 2021; Shi et al., 2022). For healthcare problem, few-shot learning is applied to improve diagnosis of uncommon diseases (Zhang et al., 2019; Suo et al., 2020; Tan et al., 2022a), drug discovery (Altae-Tran et al., 2017; Yao et al., 2021a;c; Luo et al., 2019), novel cell type classification (Brbić et al., 2020), and genomic survival analysis (Qiu et al., 2020). However, when it comes to new drug recommendations, these works fail to capture the complex relationship between diseases and drugs, and will also be largely influenced by the noisy supervision signal.

## 6    CONCLUSION

As new drugs get approved and come to the market, drug recommendation models trained on existing drugs can quickly become outdated. In this paper, we reformulate the task of drug recommendation for new drugs, which is largely ignored by prior works. We propose a meta-learning framework to solve the problem by modeling the complex relations among diseases and drugs, and eliminating the numerous false-negative patients with an external knowledge base. We evaluate EDGE on two medical datasets and show superior performance compared to all baselines. We also provided detailed analyses of ablation analyses and qualitative studies.

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
