# OpenReview forum: "Knowledge-Driven New Drug Recommendation"
_ICLR.cc/2023/Conference — Submitted to ICLR 2023_

### Official Review · Reviewer_GvBg · 2022-10-23

**Confidence:** 4
**Correctness:** 3
**Technical Novelty And Significance:** 2
**Empirical Novelty And Significance:** 3
**Recommendation:** 3

**Clarity, Quality, Novelty And Reproducibility:**

Clarity: Some details about model design and experiment setting are not clear enough, making the paper less clear. E.g., why only one negative sample is enough in prototypical network? How drug recommendation model in baselines are trained and fine-tuned? It's also suggested to provide more data statistical information to make the model design and experiment result easier to be understood, e.g., how many diseases and procedures a patient has on average?

Quality: This paper reads well. The proposed model is reasonable and the experiments are designed properly to demonstrate the effectiveness of EDGE to recommend new drugs. Ablation study is conducted to evaluate the contribution of each component.

Novelty: This paper proposes a new task, while the proposed method is not novel enough. The encoding module 'Ontology-enhanced drug representation' and 'bi-GRU' have been proposed in previous work about drug recommendation as mentioned in the paper, and prototypical network has also been applied in recommendation systems [Sankar et.al., 2021].

Reproducibility: The proposed EDGE is reproduced as the model and data process method is clear. While the setting of baselines needs to be detailed as mentioned above.

**Strength And Weaknesses:**

Strengths:
1.	Proposing a new and meaningful task of new drug recommendation.
2.	Modeling the task as a few-shot classification problem which is reasonable.

Weaknesses:
1. The application scenario is not clear. The designed experiment to answer the following question: given a new drug, which patient would take it? While in clinical application, it's hard to recommend a set of proper drugs for a specific patient -- it may take too much time to check all drugs one by one as the proposed model only take one drug as input.

2. It's not clear that, considering EDGE lacks patients' history and the few-shot learning setting, can EDGE perform well on "old" drugs compared with baselines such SafeDrug? If not, how to merge this new drug rec model with traditional "old" drug rec model to obtain a complete drug list is also a problem.

3. "Analysis on False Negatives" in sec. 4.3 seems not reasonable, as there exist lots of drug-drug interaction and drug-disease interaction. Adjusting the metrics according to MEDI without considering DDI doesn't make sense.

4. The proposed solution is composed four existing modules, which lowers the technical novelty of the work.


**Summary Of The Paper:**

The authors formulate the task of new drug recommendation as a few-shot learning task and propose a framework to solve it. They first utilize the drug ontology to learn the drug representation. Then they represent each patient as a set of phenotype vectors. Given a new drug, they make recommendations by measuring the similarity between representations of query patients and corresponding prototypes. They further introduce a knowledge-guided negative sampling strategy to eliminate false-negative effect.

**Summary Of The Review:**

New task, reasonable but less novel method, unclear application scenario. Less qualified to publish in ICLR.

---

### Official Review · Reviewer_54ZT · 2022-10-24

**Confidence:** 4
**Correctness:** 2
**Technical Novelty And Significance:** 2
**Empirical Novelty And Significance:** 2
**Recommendation:** 5

**Clarity, Quality, Novelty And Reproducibility:**

Regarding the clarity, the organization is generally ok and the writing can be further improved. For example, while I agree that the adoption of phenotype-based representation is good, related justification is not clearly presented. A bit more details regarding how to extract phenotypes from CCS will be useful.

Regarding novelty, the main contribution is more on the way of integrating existing methods, and less on fundamental ones.

**Strength And Weaknesses:**

Strengths:
+ The problem formulation is clearly presented in general.
+ The use of ontological approach to deal with the cases with a small dataset is a reasonable one. Also the way they make use of the ontology is proper and reasonable. E.g., using the knowledge for the negative sampling to handle the false-negative cases is a smart one.
+ Introducing the phenotypes can support interpretability of the model inferred as presented in the paper.

Weaknesses:
- While the problem seems interesting, it is not clear if the proposition of recommending new drugs based on only the support of a few patient record is a valid one or not. From the safety perspective, should drug be more carefully evaluated before it should be recommended? This seems to make it different from other recommendation tasks. This issue is not much discussed in the paper. This weakens the motivation of the problem and then the importance of the proposed techniques.
- The main novelty is more on the way of integrating existing methods, and less on fundamental ones.
- Only diagnosis and procedures are extracted to define the phenotypes. Works on computational phenotyping often include lab tests results. Should they be included as well? If not, what are the reasons? Related consideration is not presented in the paper.
- Regarding presentation, the organization is generally ok and the writing can be further improved. For example, while I agree that the adoption of phenotype-based representation is good, related justification is not clearly presented. A bit more details regarding how to extract phenotypes from CCS will be useful.

**Summary Of The Paper:**

The paper proposes the use of few-shot learning together with the use of medical ontologies under the prototypical network architecture for new drug recommendation. The use of few-shot learning and ontologies is motivated by the fact that data related to new drugs is limited. Several tricks are suggested, including the use of ontological representation computed using Clinical-Bert and ontology, phenotype representation based on CCS, allowing the phenotype-based representation to be drug-dependent, as well as using ontology to obtain negative samples for training the prototypical network for handling the false-negative drug issue. They tried to compare with a number of benchmark methods and reported improvement on accuracy for new drug recommendation.

**Summary Of The Review:**

Introducing various ways to integrate the ontological approaches to the address the challenges encountered in new drug recommendation is the main contribution of this paper. The contribution is not considered fundamental. While treating new drug recommendation as few-shot learning is an interesting one, the validity of recommending new drugs based on a small set of patient records related to the new drugs is unclear. This weakens the motivation of the paper, while I find this work and proposed tricks interesting.

---

### Official Review · Reviewer_usk8 · 2022-10-25

**Confidence:** 4
**Correctness:** 3
**Technical Novelty And Significance:** 3
**Empirical Novelty And Significance:** 2
**Recommendation:** 5

**Clarity, Quality, Novelty And Reproducibility:**

Clarity: The paper is well organized. Most parts of the paper are written in a clear way.

Quality: The quality of the paper can be further improved.

Novelty: The novelty is fair. But the overall contribution to the healthcare domain is limited.

Originality: Good.

Reproducibility: There are not source codes provided.


**Strength And Weaknesses:**

S1: The methodology part is written in a clear way and well addressed the main claims in the introduction part.

S2: It is an interesting idea to consider new drug recommendations as a few-shot learning problem.


W1: It's concerned to apply bi-direction GRU to the contextualized disease representations, as there are no directions among diseases. It just demonstrated better performance by using bi-directional GRG from the experiment.

W2: The proposed model does not consider patients' historical EHR data. So some facts, such as disease progression, and ongoing treatments are hard to integrate into the EDGE model, which cannot support the paper’s claim about tackling complex relations among diseases and drugs

W3: It is better to involve more components, e.g., loss functions, and data flow progress, in Figure 1 to make it clearer.


**Summary Of The Paper:**

This paper considers new drug recommendations as a few-shot learning problem.  The proposed model can quickly adapt to the recommendation for a new drug with limited prescription data from a few support patients by addressing complex relations among diseases and drugs and numerous false-negative patients. Experiments on two real-world datasets demonstrate the effectiveness of EDGE.


**Summary Of The Review:**

Below the acceptance threshold.
Please see Strength And Weaknesses for detail.

---

### Official Review · Reviewer_YURq · 2022-10-26

**Confidence:** 4
**Correctness:** 4
**Technical Novelty And Significance:** 1
**Empirical Novelty And Significance:** Not applicable
**Recommendation:** 3

**Clarity, Quality, Novelty And Reproducibility:**

- Clarity -- Yes
- Quality -- an application paper with less innovation in terms of proposed techniques
- Novelty - None
- Reproducibility -- Could not evaluate

**Strength And Weaknesses:**

Strengths
- Paper is very well written
- Uses two real-world datasets MMIC-IV and CLAIMS
- The Edge framework is compared against several well known techniques from literature such as GameNet, SafeDrug etc.

Weakness
- Relies on standard techniques proposed in literature -- thus less innovative.

**Summary Of The Paper:**

A meta-learning framework, EDGE, for new drug recommendation is proposed using relationships between drugs and diseases.

**Summary Of The Review:**

The meta-learning framework proposed using few shot learning relies primarily on existing techniques, already well studied in literature.
No new techniques are presented.

Many sophisticated algorithms exist for training -- why is a gradient descent algorithm chosen, given that it is known to be slow?

---

### Decision · Program_Chairs · 2023-01-20

**Decision:**

Reject

**Justification For Why Not Higher Score:**

The authors did not respond to the referees' comments.

**Justification For Why Not Lower Score:**

The authors did not respond to the referees' comments.

**Metareview: Summary, Strengths And Weaknesses:**

In this paper, the author(s) proposed the meta-learning approach for knowledge-driven new drug recommendation. The author(s) reasonably model the task as a few-shot classification problem. The proposed procedure was applied to two real-world datasets: MIMIC-IV and CLAIMS. The referees raised many concerns about the paper while the author(s) did not respond. In summary, there is less enthusiasm from the referees to accept the current version of this paper at the 2023 ICLR.